# Structures of TorsinA and its disease-mutant complexed with an activator reveal the molecular basis for primary dystonia

F Esra Demircioglu[1], Brian A Sosa[1], Jessica Ingram[2], Hidde L Ploegh[2], Thomas U Schwartz[1]*

[1]Department of Biology, Massachusetts Institute of Technology, Cambridge, United States; [2]Whitehead Institute for Biomedical Research, Cambridge, United States

**Abstract** The most common cause of early onset primary dystonia, a neuromuscular disease, is a glutamate deletion (ΔE) at position 302/303 of TorsinA, a AAA+ ATPase that resides in the endoplasmic reticulum. While the function of TorsinA remains elusive, the ΔE mutation is known to diminish binding of two TorsinA ATPase activators: lamina-associated protein 1 (LAP1) and its paralog, luminal domain like LAP1 (LULL1). Using a nanobody as a crystallization chaperone, we obtained a 1.4 Å crystal structure of human TorsinA in complex with LULL1. This nanobody likewise stabilized the weakened TorsinAΔE-LULL1 interaction, which enabled us to solve its structure at 1.4 Å also. A comparison of these structures shows, in atomic detail, the subtle differences in activator interactions that separate the healthy from the diseased state. This information may provide a structural platform for drug development, as a small molecule that rescues TorsinAΔE could serve as a cure for primary dystonia.

## Introduction

*For correspondence: tus@mit.edu

Torsins belong to the AAA+ (ATPases associated with a variety of cellular activities) ATPase family, a functionally diverse group of enzymes, which are fueled by ATP hydrolysis. AAA+ ATPases organize in structurally distinct fashions and interact with various accessory elements to remodel their protein or nucleic acid substrates (*Erzberger and Berger, 2006*; *Wendler et al., 2012*; *White and Lauring, 2007*). Torsins are poorly understood AAA+ proteins with yet elusive functions and unknown substrates (*Laudermilch and Schlieker, 2016*; *Rose et al., 2015*). Among the five human torsins (TorsinA, TorsinB, Torsin2A, Torsin3A and Torsin4A), neuronally expressed TorsinA carries the most clinical significance since it is at the root of primary dystonia. Primary dystonia is a devastating neuromuscular disease that is predominantly caused by the deletion of glutamate 302 or 303 (ΔE) in TorsinA (*Goodchild et al., 2005*; *Ozelius et al., 1997*). The etiology of primary dystonia is poorly understood (*Breakefield et al., 2008*; *Granata and Warner, 2010*), and there is currently no known cure for it.

TorsinA is an unusual AAA+ ATPase, because, unlike any other family member (*Erzberger and Berger, 2006*; *Laudermilch and Schlieker, 2016*; *Rose et al., 2015*; *White and Lauring, 2007*), it is localized to the endoplasmic reticulum (ER) and the contiguous perinuclear space (PNS), and because it is not self-activated, but instead needs the AAA+-like proteins Lamina-associated protein 1 (LAP1) or Luminal domain like LAP1 (LULL1) to catalyze ATP hydrolysis (*Brown et al., 2014*; *McCullough and Sundquist, 2014*; *Sosa et al., 2014*). LAP1 is a type-II transmembrane protein, which resides at the inner nuclear membrane (INM) through its association with the nuclear lamina

**eLife digest** A group of enzymes known as the AAA+ ATPase family have a wide variety of roles in the cell. They are able to break down a molecule called ATP and use the energy released to change the structure of other 'target' molecules. TorsinA is one such AAA+ ATPase and is found primarily in nerve cells inside two cell compartments called the endoplasmic reticulum and the perinuclear space. There, TorsinA interacts with one of two proteins – called LULL1 and LAP1 – that activate TorsinA. In this respect, TorsinA differs from other members of the AAA+ ATPase family, which can activate themselves without the need for additional proteins.

TorsinA and other enzymes are made up of building blocks called amino acids. Mutant forms of TorsinA that have lost a particular amino acid cause primary dystonia, an incurable neuromuscular disease. This amino acid is needed for TorsinA to interact with LULL1 and LAP1. Previous studies have revealed the 3D structure of LAP1 on its own, but the structure of TorsinA remained unknown.

One way to study the structure of enzymes is to use a technique called X-ray crystallography. The first step in this technique is to make crystals of the protein of interest. However, it has proved difficult to make crystals of TorsinA. Demircioglu et al. have addressed this problem by using X-ray crystallography to investigate the structure of TorsinA when it is bound to LULL1. The experiments used a small molecule known as a nanobody that can specifically recognize the human TorsinA enzyme. The nanobody helped TorsinA to stay attached to LULL1 and form the crystals needed for X-ray crystallography.

The 3D structures reveal how TorsinA and LULL1 interact in a high level of detail, helping to explain how TorsinA differs from other AAA+ ATPases. In addition, by comparing how normal TorsinA and the mutant form interact with LULL1, Demircioglu et al. provide more evidence that primary dystonia is likely to be caused by the improper activation of TorsinA. The subtle differences revealed by these structures could be exploited to develop new drugs to fight this disease in the future.

(*Goodchild and Dauer, 2005*). LULL1 is a LAP1 paralog, which localizes to the outer nuclear membrane (ONM) and the continuous ER, with its N-terminal portion protruding into the cytoplasm (*Goodchild and Dauer, 2005*). The structurally similar luminal domains of LAP1/LULL1 interact with TorsinA, and they provide an arginine finger to the TorsinA active site to facilitate torsin's ATP hydrolysis (*Brown et al., 2014*; *Sosa et al., 2014*). Arginine fingers are key structural motifs of AAA + ATPases because they neutralize the transition state during ATP hydrolysis (*Wendler et al., 2012*). Since torsins lack arginine fingers themselves, this activation mechanism through LAP1/LULL1 is likely critical for their function. As reported by several labs, the disease mutant TorsinA ΔE is compromised in binding to LAP1/LULL1 (*Naismith et al., 2009*; *Zhao et al., 2013*; *Zhu et al., 2010*). Clearly, this suggests that a probable cause of primary dystonia is the lack of activation of TorsinA. In line with this suggestion, LAP1 deletion shows a similar phenotype to Torsin ΔE, and contributes to disease pathology (*Kim et al., 2010*).

To investigate the molecular basis for primary dystonia as a result of the glutamate 302/303 deletion in TorsinA, we took a structural approach. We obtained high-resolution crystal structures of TorsinA as well as TorsinAΔE, each in complex with LULL1, using a nanobody as crystallization chaperone. These structures likely open a pathway toward rational, structure-based drug design against primary dystonia.

## Results

TorsinA is a catalytically inactive AAA+ ATPase (*Brown et al., 2014*; *Zhao et al., 2013*), notoriously ill-behaved *in vitro*, primarily due to its limited solubility and stability. We partially overcame these problems by stabilizing an ATP-trapped E171Q mutant of human TorsinA (residues 51–332) by co-expressing it with the luminal activation domain of human LULL1 (residues 233–470). This resulted in a better behaved heterodimeric complex (*Figure 1A*), which, however, was still recalcitrant to our crystallization efforts. To facilitate crystallization, we isolated a nanobody (VHH-BS2) from an alpaca immunized with the TorsinA$_{EQ}$-LULL1 complex. A stable, heterotrimeric complex of TorsinA$_{EQ}$-

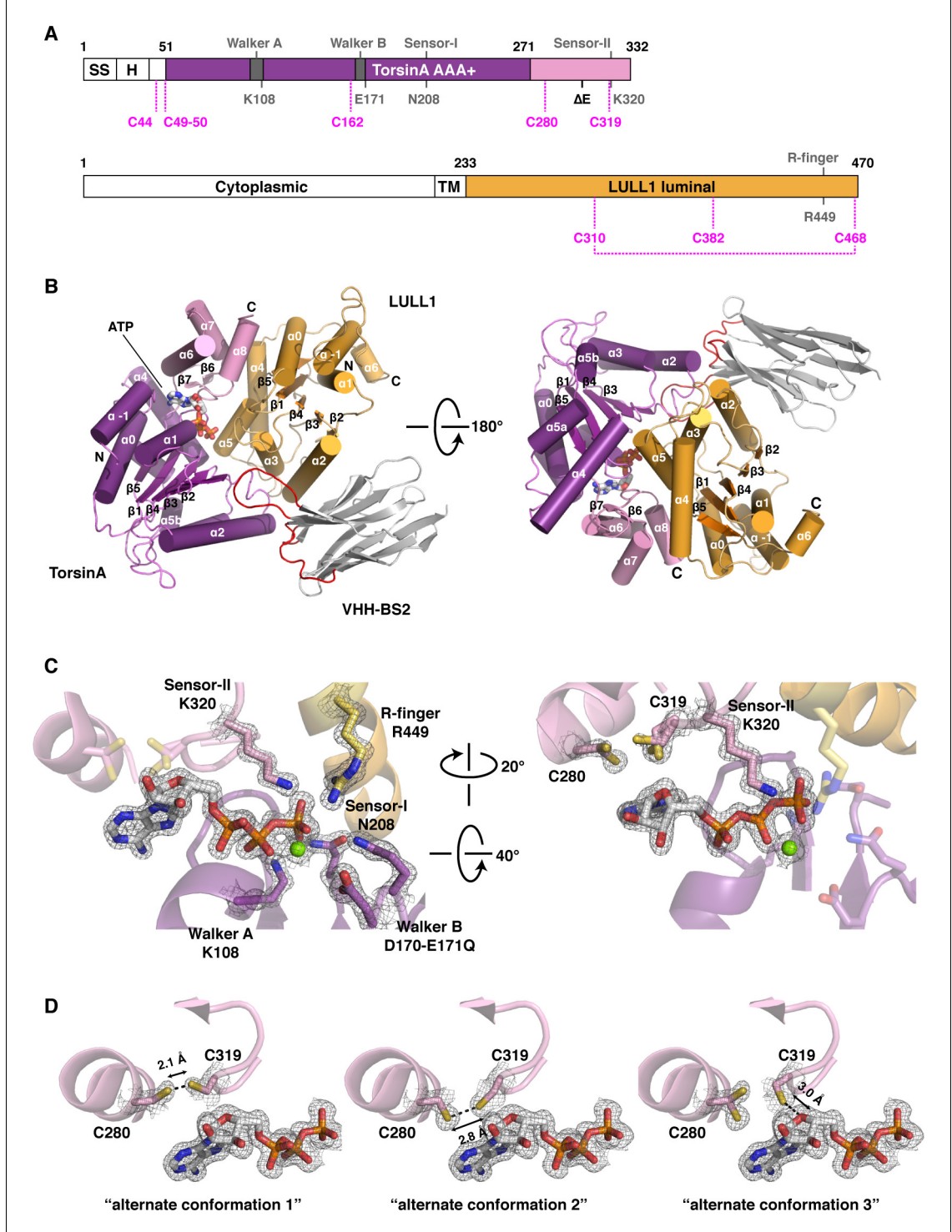

**Figure 1.** Architecture of the TorsinA-LULL1 complex. (**A**) Schematic diagrams of TorsinA and LULL1. Important residues and sequence motifs are indicated. The colored areas mark the crystallized segments. Large and small domains of TorsinA are colored in purple and pink, respectively. SS, signal sequence; H, hydrophobic region; TM, transmembrane helix. (**B**) Cartoon representation of the TorsinA-LULL1 complex in two orientations. Color-coding as in (**A**). A nanobody (VHH-BS2, grey; complementarity determining regions, red) was used as a crystallization chaperone. Numbers refer to secondary structure elements. (**C**) Close-up of the ATP binding site. Key residues are labeled. $2F_o-F_c$ electron density contoured at $2\sigma$ displayed as grey mesh. (**D**) Close-up of the proximal cysteines 280 and 319 next to the adenine base of the bound ATP. $2F_o-F_c$ electron density is contoured at $1\sigma$. The cysteine pair adopts three alternate conformations, but remains reduced in all of them.

*Figure 1 continued on next page*

*Figure 1 continued*

The following figure supplements are available for figure 1:

**Figure supplement 1.** Structural comparisons.
**Figure supplement 2.** Phylogenetic analysis of Torsins.
**Figure supplement 3.** Phylogenetic analysis of LAP1/LULL1.
**Figure supplement 4.** Nanobody interaction.
**Figure supplement 5.** Comparison of sequence motifs of AAA+ ATPases.

LULL1-VHH-BS2 was readily crystallized in the presence of ATP. We collected a 1.4 Å dataset and solved the structure by molecular replacement, using the LULL1-homolog LAP1 and a VHH template as search models (*Sosa et al., 2014*) (Materials and methods, *Table 1*). TorsinA$_{EQ}$ adopts a typical AAA+ ATPase fold (*Figure 1B*, *Figure 1—figure supplement 1*). The N-terminal nucleotide-binding or large domain (residues 55–271) is composed of a central five-stranded, parallel β-sheet surrounded by 8 α-helices. A small three-helix bundle at its C-terminus (residues 272–332), forms critical contacts with LULL1. An ATP molecule is bound in the manner characteristic of P-loop NTPases (*Wendler et al., 2012*). The Walker A and B motifs are positioned to mediate the requisite nucleotide interactions, with sensor 1 and sensor 2 regions sensing the γ-phosphate and thus the nucleotide state (*Figure 1C*). The luminal LULL1 activation domain (residues 236–470) adopts an AAA+-like conformation, very similar to its paralog LAP1 (rmsd 1.04 Å over 211 Cα positions, *Figure 1—figure supplement 1*). The AAA+-like domain comprises a central β-sheet embedded within six α-helices (*Figure 1B*). A C-terminal small domain is not present. Similar to LAP1, an intramolecular disulfide bond forms at the C terminus of LULL1, between conserved residues C310 and C468 (*Figure 1—figure supplements 1,3*). Characteristically, LULL1 lacks nucleotide binding due to the absence of Walker A and B motifs (*Sosa et al., 2014*). LULL1 forms a composite nucleotide-binding site with TorsinA by providing arginine residue 449 ('arginine finger') at the base of helix α5 (*Figure 1C*). The arginine finger activates ATP hydrolysis by TorsinA (*Brown et al., 2014*; *Sosa et al., 2014*). The small domain of TorsinA, including helix α7 featuring glutamates 302 and 303, is intimately involved in LULL1 binding. Nanobody VHH-BS2 binds both TorsinA and LULL1 at a shallow groove (*Figure 1B*, *Figure 1—figure supplement 4*). Nanobodies contain three complementarity determining regions (CDRs), with CDR3 most often making critical contacts with the antigen (*Muyldermans, 2013*). Indeed, the long CDR3 of VHH-BS2 (residues 97–112) is the main binding element in the complex.

AAA+ ATPases are organized into a number of structurally defined clades (*Erzberger and Berger, 2006*; *Iyer et al., 2004*), distinguished by shared structural elements. Comparison with other AAA+ ATPase structures shows that TorsinA fits best into a clade that also contains the bacterial proteins HslU, ClpA/B, ClpX, and Lon (HCLR clade), all of which are involved in protein degradation or remodeling (*Erzberger and Berger, 2006*). These AAA+ family members share a β-hairpin insertion that precedes the sensor-I region (*Figure 1—figure supplement 1*). TorsinA also contains this structural element, but it adopts a distinctly different orientation compared to other members of the clade; however, the pre-sensor I region may be affected by crystal packing in our structure. Two other distinct regions are present. The protein degrading or remodeling AAA+ ATPases all form hexameric rings with a central pore (*Hanson and Whiteheart, 2005*; *Olivares et al., 2016*; *White and Lauring, 2007*). 'Pore loops' in each subunit, conserved elements positioned between strand β2 and helix α2, are critical for threading the protein substrates through the ring (*Sauer and Baker, 2011*). Torsins are devoid of a pore loop consensus motif (*Figure 1—figure supplements 2,5*). TorsinA has two cysteines (Cys280, and Cys 319, which is part of the sensor-II motif), positioned near the adenine base of the ATP molecule (*Figure 1D*). These cysteines do not form a disulfide bridge in our structure. However, the conservation of Cys280 and the Gly-Cys-Lys sensor-II motif at position 318–320 (*Figure 1—figure supplements 2,5*) indicates an important functional role. A

**Table 1.** X-ray data collection and refinement statistics.

| | TorsinA-LULL1$_{233-470}$ | TorsinA$\Delta$E-LULL1$_{233-470}$ |
|---|---|---|
| PDB Code | 5J1S | 5J1T |
| **Data collection** | | |
| Space group | P2$_1$2$_1$2$_1$ | P2$_1$2$_1$2$_1$ |
| Cell dimensions | | |
| a, b, c (Å) | 75.7, 90.7, 105.1 | 75.4, 88.4, 105.3 |
| α, β, γ (°) | 90.0, 90.0, 90.0 | 90.0, 90.0, 90.0 |
| Resolution (Å) | 61–1.40 (1.45–1.40)* | 68–1.40 (1.45–1.40) |
| $R_{sym}$ | 0.06 (0.88) | 0.10 (1.98) |
| $R_{pim}$ | 0.03 (0.43) | 0.03 (0.60) |
| I / σ | 33.0 (1.5) | 30.8 (1.3) |
| Completeness (%) | 94.7 (67.5) | 97.9 (96.5) |
| Redundancy | 5.7 (4.4) | 12.4 (11.3) |
| CC(1/2) | 1.00 (0.65) | 1.00 (0.58) |
| **Refinement** | | |
| Resolution (Å) | 61.4–1.40 | 67.7–1.40 |
| No. reflections | 132956 | 134333 |
| $R_{work}$ / $R_{free}$ | 0.143/0.188 | 0.148/0.177 |
| No. atoms | 5898 | 5927 |
| Protein | 5241 | 5244 |
| Ligand/ion | 35 | 47 |
| Water | 622 | 636 |
| B factors (Å$^2$) | | |
| Protein | 31.3 | 24.0 |
| Ligand/ion | 23.2 | 17.2 |
| Water | 43.1 | 33.6 |
| r.m.s. deviations | | |
| Bond lengths (Å) | 0.014 | 0.017 |
| Bond angles (°) | 1.25 | 1.71 |
| Ramachandran | | |
| Favored/allowed/outliers (%) | 98.0/1.7/0.0 | 98.6/1.4/0.0 |

*Values in parentheses are for highest-resolution shell. One crystal was used for each dataset.

redox activity as part of the ATPase cycle therefore seems highly likely, as has been previously speculated (*Zhu et al., 2008*, *2010*).

The interaction of TorsinA with its ATPase activators LULL1 and LAP1 is of particular importance, as a prominent mutation causing primary dystonia–the deletion of glutamate 302 or 303–weakens these interactions (*Naismith et al., 2009*; *Zhao et al., 2013*; *Zhu et al., 2010*). But why and how? The TorsinA-LULL1 interface extends over an area of 1439 Å$^2$. The main structural elements involved in this interaction are the nucleotide-binding region as well as the small domain of TorsinA, and helices α0, α2, α4 and α5 of LULL1 (*Figure 1*, *Figure 1—figure supplements 2,3*, *Figure 2A*). The exact position of the small domain of TorsinA relative to the large domain is likely dictated by the sensor II motif, preceding α8, which directly contacts the γ-phosphate of ATP through Lys 320, thus serving as an anchor point. A switch to ADP presumably weakens this connection, such that the small domain would become more loosely attached to the large domain. This could explain the observed ATP-dependency of LAP1/LULL1 binding (*Goodchild and Dauer, 2005*; *Naismith et al., 2009*;

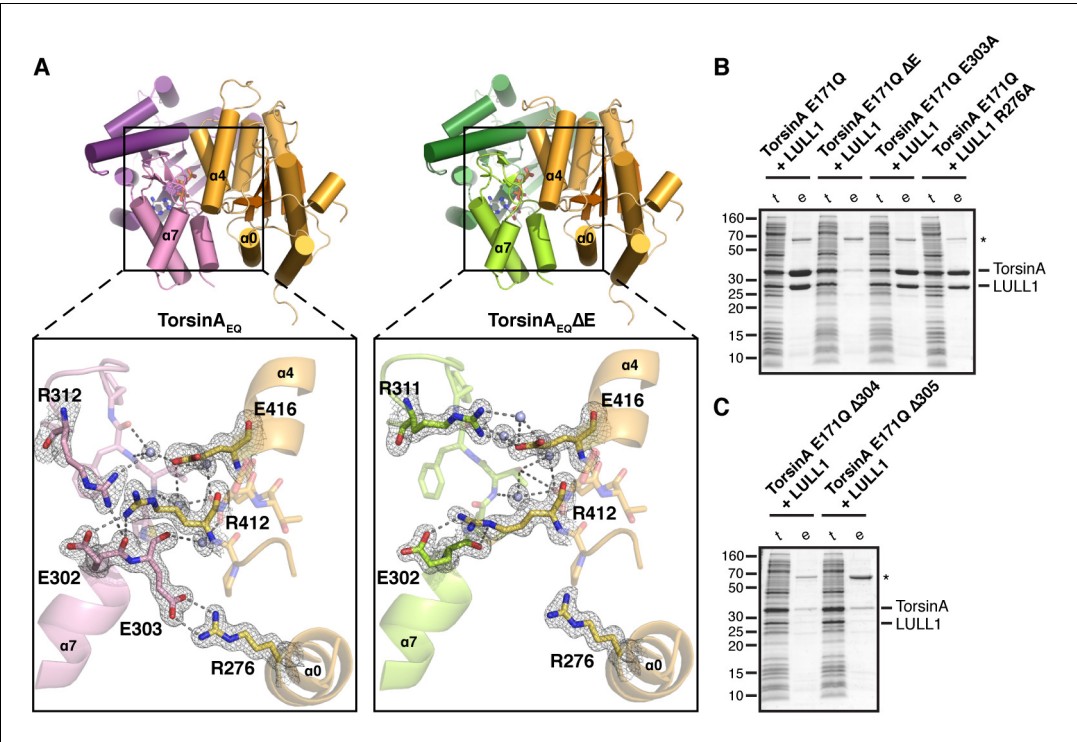

**Figure 2.** Analysis of the TorsinA-LULL1 interface. (A) Side-by-side comparison of TorsinA-ATP-LULL1 (left) and TorsinAΔE-ATP-LULL1 (right). Zoomed insets show the atomic details of the interactions between TorsinA/TorsinAΔE and LULL1, with a focus on the ΔE303 area. (B and C) Mutational analysis of the TorsinA-LULL1 interface. Substitution or deletion of residues involved in TorsinA-LULL1 binding were probed using a Ni-affinity co-purification assay with recombinant, bacterial-expressed protein. Only TorsinA is His-tagged. SDS-PAGE analysis is shown. Lack of binding is observed by the absence of complex (uncomplexed His-tagged TorsinA is insoluble). t, total lysate, e, Ni eluate. Asterisk denotes an unrelated contaminant.

The following figure supplement is available for figure 2:

**Figure supplement 1.** Structural mapping of mutations causing dystonia.

Zhao et al., 2013; Zhu et al., 2010). Within the small domain, helix α7, the following loop, and the terminal helix α8 contain all the critical residues. Glutamate 302 and 303 are positioned at the very end of helix α7, and both are involved in TorsinA contacts. Specifically, Glu 303 forms a prominent charge interaction with Arg 276 of LULL1. TorsinA Lys113 – LULL1 Glu385, TorsinA Asp316 - LULL1 Arg419, TorsinA Lys317 - LULL1 Glu415, TorsinA Asp327 - LULL1 Lys283 are additional charge interactions.

To investigate the atomic details of the weakened binding of TorsinAΔE to LAP1/LULL1, and thus the molecular basis of primary dystonia, we made use of the observation that VHH-BS2 also stabilizes the TorsinA_EQΔE(ATP)-LULL1 interaction. We were able to crystallize TorsinA_EQΔE(ATP)-LULL1-VHH-BS2 and determine its structure at a resolution of 1.4 Å. Not surprisingly, the overall structure is almost identical to the wild-type protein (0.34 Å rmsd over 276 Cα atoms for TorsinA, 0.27 Å rmsd over 226 Cα atoms for LULL1), except for critical differences in the TorsinA-LULL1 interface (*Figure 2A*). The principal difference is that helix α7 is shortened due to the missing Glu 303, with a slight–but significant–restructuring of the loop that follows to establish the connection with helix α8. For future reference, we suggest renaming the ΔE mutation ΔE303, rather than ΔE302/303, since the position of Glu 302 is effectively unchanged. In the dystonia mutant, the TorsinA Glu 303 – LULL1 Arg 276 charge interaction is lost, and the hydrogen-bonding network involving TorsinA Glu 302, Phe 306 and Arg312, as well as LULL1 Arg412 and Glu416 is disrupted (*Figure 2A*). To determine the importance of different TorsinA residues for LULL1 binding, we performed a co-purification assay

**Table 2.** Dystonia mutations.

| Protein | Mutation | Structural consequence | Reference |
|---|---|---|---|
| TorsinA | ΔE302/303 | Weakened LAP1/LULL1 binding | (*Ozelius et al., 1997*) |
| TorsinA | ΔF323-Y328 | Weakened LAP1/LULL1 binding | (*Leung et al., 2001*) |
| TorsinA | R288Q | Weakened LAP1/LULL1 binding | (*Zirn et al., 2008*) |
| TorsinA | F205I | Folding problem | (*Calakos et al., 2010*) |
| TorsinA | D194V | Change to the conserved, noncatalytic interface | (*Cheng et al., 2014*) |
| TorsinA | ΔA14-P15 | Improper cellular targeting | (*Vulinovic et al., 2014*) |
| TorsinA | E121K | Charge inversion at the membrane proximal interface | (*Vulinovic et al., 2014*) |
| TorsinA | V129I | Folding problem | (*Dobričić et al., 2015*) |
| TorsinA | D216H (modifier) | Surface change; consequence unclear | (*Kamm et al., 2008*; *Kock et al., 2006*) |
| LAP1 | c.186deiG (p.E62fsTer25) | Lack of the luminal activation domain of LAP1 | (*Kayman-Kurekci et al., 2014*) |
| LAP1 | E482A* | Improper folding | (*Dorboz et al., 2014*) |

*Assesment based on the equivalent residue in LULL1 (E368).

(*Figure 2B,C*). His-tagged, ATP-trapped TorsinA$_{EQ}$ (residues 51–332) and mutants thereof were recombinantly co-expressed with LULL1 (residues 233–470), but without VHH-BS2, in bacteria. Binding was tested in a co-purification assay using Ni-affinity. The Torsin$_{EQ}$ΔE303 mutation abolishes binding in this assay, as expected (*Figure 2B*). Since unbound TorsinA$_{EQ}$ is largely insoluble, absence of binding is not registered as an appearance of TorsinA$_{EQ}$ alone, but rather as a lack of eluted protein complex altogether. Eliminating the salt bridge between TorsinA Glu303 and LULL1 Arg276 does not disrupt the TorsinA-LULL1 interaction (*Figure 2B*). However, ΔMet304 and ΔThr305 both phenocopy ΔE303 in abolishing LULL1 binding (*Figure 2C*). This is in full agreement with published *in vivo* data using similar mutants (*Goodchild and Dauer, 2004*). The intricate network of interactions of the α7-α8 loop of TorsinA is crucial for LULL1 binding. Since the ΔE mutation causes a local change within the small domain of TorsinA rather than protein misfolding, it may be possible to rescue binding by developing a small molecule that resurrects the weakened TorsinAΔE-LAP1/LULL1 interaction.

Although TorsinAΔE303 is the most prevalent mutation that causes primary dystonia, it is not the only one (*Laudermilch and Schlieker, 2016*; *Rose et al., 2015*). We examined the structural consequence of all known mutations (*Figure 2—figure supplement 1*, *Table 2*). Based on our structural data, we strongly predict that most mutations likely cause protein misfolding or they weaken or abolish LAP1/LULL1 binding. Conversely, the two dystonia-mutations found in LAP1 presumably affect torsin interaction. Our structural data, therefore, clearly support the hypothesis that improper torsin activation is the likely cause of primary dystonia (*Kim et al., 2010*).

## Discussion

The biological function of TorsinA remains enigmatic (*Granata et al., 2011*; *Jokhi et al., 2013*; *Liang et al., 2014*; *Nery et al., 2008*, *2011*). Because TorsinA belongs to the AAA+ ATPase superfamily, with specific homology to the bacterial proteins HslU, ClpX, ClpA/B and Lon, it is generally assumed that TorsinA is involved in protein remodeling or protein degradation (*Laudermilch and Schlieker, 2016*; *Rose et al., 2015*). However, a substrate of TorsinA has yet to be identified.

The TorsinA structure enables a more thorough comparison to other AAA+ ATPases, particularly with regard to the functionally relevant oligomerization state. After the discovery that LAP1/LULL1 are Arg-finger containing TorsinA activators with a AAA+-like structure, it seemed reasonable to suggest that TorsinA and LAP1/LULL1 likely form heterohexameric rings ((TorsinA-ATP-LAP1/LULL1)$_3$) in order to function (*Brown et al., 2014*; *Sosa et al., 2014*). However, the predominant oligomeric form of recombinant TorsinA-ATP-LAP1/LULL1 complex *in vitro* and in solution is the heterodimer (*Brown et al., 2014*; *Sosa et al., 2014*). In addition, torsin variants have been reported to occur in various oligomeric forms as detected by Blue Native PAGE (BN-PAGE) (*Goodchild et al.,*

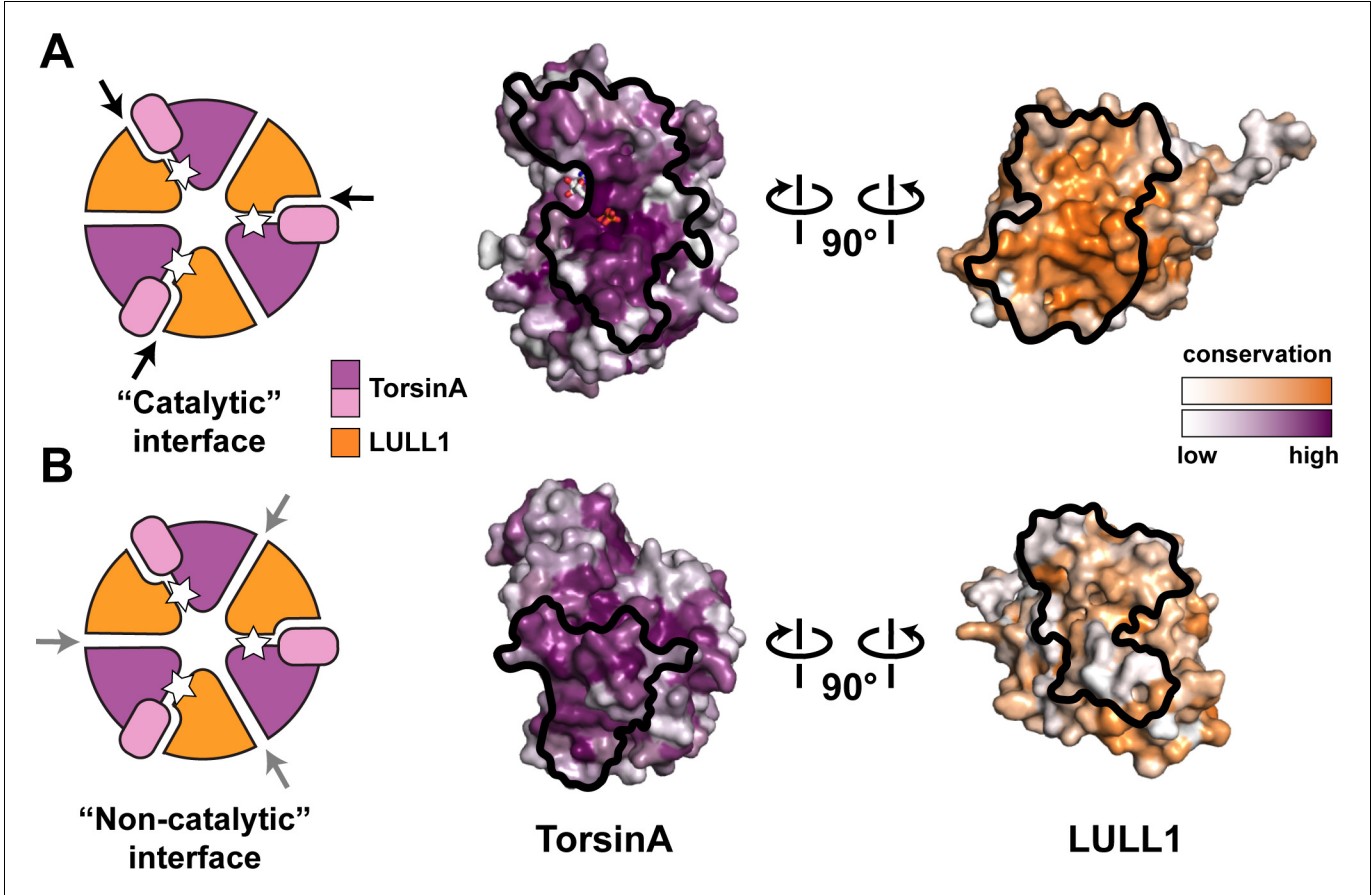

**Figure 3.** Oligomerization of TorsinA-LULL1. (**A**) Left, Schematic representation of a hypothetical heterohexameric (TorsinA-LULL1)₃ ring model, in analogy to canonical AAA+ ATPases. White star represents ATP. Since LULL1 cannot bind a nucleotide, there would be three catalytic (nucleotide-bound) and three non-catalytic interfaces per ring. Open-book representation of the catalytic interface between TorsinA and LULL1, as seen in this study. Black line marks the outline of the interface. Color gradient marks conservation across diverse eukaryotes. (**B**) The same analysis as in (**A**), but for the hypothetical 'non-catalytic' interface. The interface model on the right is based on swapping the TorsinA and LULL1 positions in the TorsinA-LULL1 complex.

*2015*; *Jungwirth et al., 2010*; *Vander Heyden et al., 2009*). Our structure now raises doubts about the physiological relevance of a heterohexameric ring (*Figure 3*). First, we note that the small domain of TorsinA is essential for LAP1/LULL1 binding (*Figure 2C*). This is reminiscent of the related HCLR AAA+ clade members where the small domain is known to be critical for hexamerization (*Bochtler et al., 2000*; *Mogk et al., 2003*). The importance of the small domain for oligomerization in the context of torsins has also been discussed recently (*Rose et al., 2015*). Neither LAP1 nor LULL1 harbor a small domain, arguing against formation of a stable heteromeric ring, or, alternatively, suggesting a ring of substantially different architecture. Second, ring formation is important for AAA+ ATPases that thread their protein substrate through a central pore for refolding or for degradation. This central pore is lined with conserved 'pore loops' that are essential for function (*White and Lauring, 2007*). Neither TorsinA and its homologs, nor LAP1/LULL1 have 'pore loop' equivalents (*Figure 1—figure supplement 5*). TorsinA is therefore unlikely to actually employ a peptide threading mechanism that involves a central pore. Third, the surface conservation of LAP1/LULL1 also argues against a heteromeric ring assembly. Although the catalytic, ATP-containing interface with TorsinA is well-conserved, the presumptive non-catalytic, nucleotide-free interface is not (*Figure 3B*). Importantly and in contrast to LAP1/LULL1, the same analysis for TorsinA shows that its 'backside' is conserved. TorsinA may therefore interact in homotypic fashion with TorsinA, with other torsin homologs, or even with an additional, yet unidentified protein. This could mean that the

previously observed hexameric assemblies (*Goodchild et al., 2015*; *Jungwirth et al., 2010*; *Sosa et al., 2014*) may only contain one LAP1/LULL1 unit, and multiple torsin units, a property that the employed assays would not have differentiated. It is also possible, that the reported hexameric assemblies reflect a vestigial, yet physiologically irrelevant property, perhaps just of the evolutionary origin of the Torsin-LAP1/LULL1 system. Taking all the existing data into account, it is suggestive that TorsinA may be an exceptional AAA+ ATPase in that it simply acts as a heterodimer, together with LAP1 or LULL1 functioning as an activator. As long as the biological function and the substrate for TorsinA are unclear, however, the physiologically relevant oligomeric state of TorsinA ultimately remains a matter of speculation. Given the unique properties of TorsinA, keeping an open mind about TorsinA assembly into its functional state is called for, as it may well differ more than anticipated from well-studied AAA+ ATPase systems.

The observation that the nanobody VHH-BS2 stabilizes the TorsinAΔE303-LULL1 suggests that it could possibly be used directly as a therapeutic. After all, it could directly rescue TorsinA activity. There are, however, at least two major problems. First, VHH-BS2 only recognizes the TorsinA- (or TorsinAΔE303-) LULL1 complex, but not the homologous TorsinA-LAP1 complex. The function of LULL1 is still poorly understood, but a knockdown does not generate an NE blebbing phenotype (*Goodchild et al., 2015*; *Turner et al., 2015*; *Vander Heyden et al., 2009*), which is symptomatic for a TorsinA knockout (*Goodchild et al., 2005*) or a LAP1 knockdown (*Kim et al., 2010*). Therefore, resurrecting activation of TorsinAΔE303 via LULL1 is unlikely to ameliorate the dystonia phenotype. Furthermore, the nanobody interaction site on the TorsinA-LULL1 interface is very likely oriented toward the ER membrane, which can be inferred from the relative positions of the membrane anchor of LULL1 and the hydrophobic, likely membrane-proximal N-terminal region of TorsinA. These topological restraints suggest that the nanobody will not bind *in vivo*, but that it is of significant use for *in vitro* studies.

## Materials and methods

### Constructs, protein expression and purification

DNA sequences encoding human TorsinA (residues 51–332) and the luminal domain of human LULL1 (residues 233–470) were cloned into a modified ampicillin resistant pETDuet-1 vector (EMD Millipore). TorsinA, N-terminally fused with a human rhinovirus 3C protease cleavable 10xHis-7xArg tag, was inserted into the first multiple cloning site (MCS), whereas the untagged LULL1 was inserted into the second MCS. Mutations on TorsinA and LULL1 were introduced by site-directed mutagenesis. The untagged VHH-BS2 nanobody was cloned into a separate, modified kanamycin resistant pET-30b(+) vector (EMD Biosciences).

To co-express TorsinA (EQ or EQ/ΔE), LULL1 and VHH-BS2 for crystallization, the *E. coli* strain LOBSTR(DE3) RIL (Kerafast, Boston MA) (*Andersen et al., 2013*) was co-transformed with the two constructs described above. Cells were grown at 37°C in lysogeny broth (LB) medium supplemented with 100 µg ml$^{-1}$ ampicillin, 25µg ml$^{-1}$ kanamaycin and 34 µg ml$^{-1}$ chloramphenicol until an optical density (OD$_{600}$) of 0.6–0.8 was reached, shifted to 18°C for 20 min, and induced overnight at 18°C with 0.2 mM isopropyl β-D-1-thiogalactopyranoside (IPTG). The bacterial cultures were harvested by centrifugation, suspended in lysis buffer (50 mM HEPES/NaOH pH 8.0, 400 mM NaCl, 40 mM imidazole, 10 mM MgCl$_2$, and 1 mM ATP) and lysed with a cell disruptor (Constant Systems). The lysate was immediately mixed with 0.1 M phenylmethanesulfonyl fluoride (PMSF) (50 µl per 10 ml lysate) and 250 units of TurboNuclease (Eton Bioscience), and cleared by centrifugation. The soluble fraction was gently mixed with Ni-Sepharose 6 Fast Flow (GE Healthcare) resin for 30 min at 4°C. After washing with the lysis buffer, bound protein was eluted in elution buffer (10 mM HEPES/NaOH pH 8.0, 150 mM NaCl, 300 mM imidazole, 10 mM MgCl$_2$, and 1 mM ATP). The eluted protein complex was immediately purified by size exclusion chromatography on a Superdex S200 column (GE Healthcare) equilibrated in running buffer (10 mM HEPES/NaOH pH 8.0, 150 mM NaCl, 10 mM MgCl$_2$, and 0.5 mM ATP). Following the tag removal by 10xHis-7xArg-3C protease, the fusion tags and the protease were separated from the complex by cation-exchange chromatography on a HiTrapS column (GE Healthcare) using a linear NaCl gradient. The flow-through from the cation-exchange chromatography, containing the protein complex, was purified again by size exclusion chromatography on a Superdex S200 column as at the previous step.

For the non-structural analysis of TorsinA and LULL1 variants, the pETDuet-1-based expression plasmid was transformed into LOBSTR(DE3) RIL cells without co-expressing nanobody VHH-BS2. $Ni^{2+}$-affinity purification was performed as described above and bound protein was eluted. Aliquots from the $Ni^{2+}$-eluate and the total lysate were collected and analyzed by SDS-PAGE gel electrophoresis.

## Crystallization

Purified TorsinA$_{EQ}$-LULL1-VHH-BS2 and TorsinA$_{EQ}\Delta$E-LULL1-VHH-BS2 complexes were concentrated up to 4–4.5 mg/ml and supplemented with 2 mM ATP prior to crystallization. The TorsinA$_{EQ}$ containing complex crystallized in 13% (w/v) polyethylene glycol (PEG) 6000, 5% (v/v) 2-Methyl-2,4-pentanediol, and 0.1 M MES pH 6.5. The TorsinA$_{EQ}\Delta$E containing complex crystallized in 19% (w/v) PEG 3350, 0.2 M AmSO$_4$, and 0.1 M Bis-Tris-HCl pH 6.5. Crystals of both complexes grew at 18°C in hanging drops containing 1 µl of protein and 1 µl of mother liquor. Clusters of diffraction quality, rod-shaped crystals formed within 3–5 days. Single crystals were briefly soaked in mother liquor supplemented with 20% (v/v) glycerol for cryoprotection and flash-frozen in liquid nitrogen.

## Data collection and structure determination

X-ray data were collected at NE-CAT beamline 24-ID-C at Argonne National Laboratory. Data reduction was performed with the HKL2000 package (*Otwinowski and Minor, 1997*), and all subsequent data-processing steps were carried out using programs provided through SBGrid (*Morin et al., 2013*). The structure of the TorsinA$_{EQ}$-LULL1-VHH-BS2 complex was solved by molecular replacement (MR) using the Phaser-MR tool from the PHENIX suite (*Adams et al., 2010*). A three-part MR solution was easily obtained using a sequential search for models of LULL1, VHH-BS2, and TorsinA. The LULL1 model was generated based on the published human LAP1 structure (PDB 4TVS, chain A), using the Sculptor utility of the PHENIX suite (LULL1$_{241–470}$ and LAP1$_{356–583}$ share 64% sequence identity). The VHH-BS2 model was based on VHH-BS1 (PDB 4TVS, chain a) after removing the complementarity determining regions (CDRs). The poly-Ala model of TorsinA was generated based on *E. coli* ClpA (PDB 1R6B) using the MODELLER tool of the HHpred server (*Söding et al., 2005*). The asymmetric unit contains one TorsinA$_{EQ}$-LULL1-VHH-BS2 complex. Iterative model building and refinement steps gradually improved the electron density maps and the model statistics. The stereochemical quality of the final model was validated by Molprobity (*Chen et al., 2010*). TorsinA$_{EQ}\Delta$E-LULL1-VHH-BS2 crystallized in the same unit cell. Model building was carried starting from a truncated TorsinA$_{EQ}$-LULL1-VHH-BS2 structure. All manual model building steps were carried out with Coot (*Emsley et al., 2010*), and *phenix.refine* was used for iterative refinement. Two alternate conformations of a loop in LULL1 (residues 428–438) were detected in the $F_o-F_c$ difference electron density maps of both structures, and they were partially built. For comparison, the cysteine residues of TorsinA at the catalytic site (residues 280 and 319 in the TorsinA$_{EQ}$ structure) were built in the reduced and the oxidized states, respectively. Building them as oxidized, disulfide-bridged residues consistently produced substantial residual $F_o-F_c$ difference density, which disappeared assuming a reduced state. Statistical parameters of data collection and refinement are all given in *Table 1*. Structure figures were created in PyMOL (Schrödinger LLC).

## Bioinformatic analysis

Torsin and LAP1/LULL1 sequences were obtained via PSI-BLAST (*Altschul et al., 1997*) and Backphyre searches (*Kelley and Sternberg, 2009*). Transmembrane domains were predicted using the HMMTOP tool (*Tusnády and Simon, 2001*). LAP1/LULL1 proteins were distinguished based on the calculated isoelectric point (pI) of their extra-luminal portions. The intranuclear domain of LAP1 has a characteristically high pI of ~8.5–10 due to a clustering of basic residues, while the cytoplasmic domain of LULL1 is distinctively more acidic. Multiple sequence alignments were performed using MUSCLE (*Edgar, 2004*), and visualized by Jalview (*Waterhouse et al., 2009*). To illustrate evolutionary conservation on TorsinA and LULL1 surfaces, conservation scores for each residue were calculated using the ConSurf server with default parameters (*Glaser et al., 2003*).

The sequences, which were used to generate the multiple sequence alignments, were also used for preparing the sequence logos of Torsins and LAP1/LULL1 in *Figure 1—figure supplement 5*. To obtain the sequence logo of the HCLR clade AAA+ ATPases, *Escherichia coli* ClpA-D2 (residues

458–758), *Escherichia coli* ClpB-D2 (residues 568–857), *Bacillus subtilis* ClpE-D2 (residues 409–699), *Saccharomyces cerevisiae* Hsp104-D2 (residues 578–868), *Escherichia coli* HslU (residues 13–443), *Bacillus subtilis* HslU (residues 15–455), *Streptomyces coelicolor* ClpX (residues 71–409), *Drosophila melanogaster* ClpX (residues 199–634), *Escherichia coli* Lon (residues 320–580), *Caenorhabditis elegans* Lon (residues 476–771), *Thermus thermophilus* ClpB-D2 (residues 536–845), *Escherichia coli* ClpX (residues 64–403), *Helicobacter pylori* ClpX (residues 77–430), *Haemophilus influenza* HslU (1–444), *Bacillus subtilis* Lon (residues 300–590), *Bacillus subtilis* ClpC-D2 (residues 486–802), *Saccharomyces cerevisiae* Hsp78-D2 (residues 482–794) and *Arabidopsis thaliana* Hsp101-D2 (residues 547–849) sequences were used. All sequence logos were generated using WebLogo (*Crooks et al., 2004*).

## Generation and selection of nanobodies

The purified human TorsinA$_{EQ}$-LULL1 complex was injected into a male alpaca (*Lama pacos*) for immunization. Generation and screening of nanobodies was carried out as previously described (*Sosa et al., 2014*). Each of the selected nanobodies was subcloned into a pET-30b(+) vector with a C-terminal His$_6$-tag. Each nanobody was bacterially expressed and Ni$^{2+}$-affinity purified essentially as described (see above). Different from the TorsinA-containing preparations, MgCl$_2$ and ATP were eliminated from all buffer solutions. The Ni$^{2+}$-eluate was purified via size exclusion chromatography on a Superdex S75 column (GE Healthcare) in running buffer (10 mM HEPES/NaOH pH 8.0, 150 mM NaCl). Nanobody binding was validated by size exclusion chromatography on a 10/300 Superdex S200 column in 10 mM HEPES/NaOH pH 8.0, 150 mM NaCl, 10 mM MgCl$_2$ and 0.5 mM ATP. Equimolar amounts of TorsinA$_{EQ}$-LULL1 and TorsinA$_{EQ}$-LULL1-VHH were loaded and nanobody binding was monitored by a shift in the elution profile and via SDS-PAGE analysis. After validating VHH-BS2 interaction with TorsinA$_{EQ}$-LULL1, the C-terminal His$_6$-tag of VHH-BS2 was removed from the pET-30b(+) vector for co-purification experiments.

## Acknowledgements

The authors thank Ulrike Kutay for many helpful discussions pertinent to this study. The work in the Schwartz lab was supported by the Foundation for Dystonia Research and a National Institutes of Health grant (AR065484). Work in the lab of HLP was supported by an NIH Director's Pioneer award. Structural data are based upon research conducted at the Northeastern Collaborative Access Team beamlines, which are funded by the National Institute of General Medical Sciences from the National Institutes of Health (P41 GM103403). This research used resources of the Advanced Photon Source, a US Department of Energy (DOE) Office of Science User Facility operated for the DOE Office of Science by Argonne National Laboratory under Contract No. DE-AC02-06CH11357.

## Additional information

### Competing interests

FED, BAS and TUS: Filed a provisional patent application protecting the use of the crystal structures (U.S.P.T.O. No. 62/330,683). The other authors declare that no competing interests exist.

### Funding

| Funder | Grant reference number | Author |
| --- | --- | --- |
| National Institutes of Health | AR065484 | Hidde L Ploegh<br>Thomas U Schwartz |
| National Institutes of Health | Director's Pioneer award | Hidde L Ploegh |
| Foundation for Dystonia Research | | Thomas U Schwartz |
| National Institute of General Medical Sciences | P41 GM103403 | Thomas U Schwartz |

The funders had no role in study design, data collection and interpretation, or the decision to submit the work for publication.

## Author contributions

FED, Conception and design, Acquisition of data, Analysis and interpretation of data, Drafting or revising the article; BAS, Acquisition of data, Analysis and interpretation of data; JI, Acquisition of data, Drafting or revising the article; HLP, Analysis and interpretation of data, Drafting or revising the article; TUS, Conception and design, Analysis and interpretation of data, Drafting or revising the article

## Author ORCIDs

F Esra Demircioglu, http://orcid.org/0000-0002-3866-2742
Thomas U Schwartz, http://orcid.org/0000-0001-8012-1512

# Additional files

## Major datasets

The following datasets were generated:

| Author(s) | Year | Dataset title | Dataset URL | Database, license, and accessibility information |
| --- | --- | --- | --- | --- |
| F Esra Demircioglu, Thomas U Schwartz | 2016 | TorsinA-LULL1 complex, H. sapiens, bound to VHH-BS2 | http://www.rcsb.org/pdb/explore/explore.do?structureId=5J1S | Publicly available at RCSB Protein Data Bank (accession no. 5J1S) |
| F Esra Demircioglu, Thomas U Schwartz | 2016 | TorsinAdeltaE-LULL1 complex, H. sapiens, bound to VHH-BS2 | http://www.rcsb.org/pdb/explore/explore.do?structureId=5J1T | Publicly available at RCSB Protein Data Bank (accession no. 5J1T) |

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
