## [Decision Letter]

Thank you for submitting your article "Structures of TorsinA and its dystonia-mutant in complex with an activator reveal the molecular basis for the disorder" for consideration by *eLife*. Your article has been favorably evaluated by John Kuriyan (Senior editor) and three reviewers, one of whom, Wesley I Sundquist (Reviewer #1), is a member of our Board of Reviewing Editors, and another is Christopher Hill (Reviewer #3).

The reviewers have discussed the reviews with one another and the Reviewing Editor has drafted this decision to help you prepare a revised submission.

The manuscript by Demircioglu et al. describes high resolution crystal structures of: 1) an "E/Q" (enzymatically inactive), but otherwise wild type TorsinA protein, in complex with its activator, LULL1, and 2) an analogous structure of the dystonia-causing "delta E" TorsinA protein in complex with LULL1. Strengths of the manuscript include the medical significance and scientific interest in TorsinA, and the importance for the field of knowing the precise structure of TorsinA and how it interacts with its activators. Two potentially important mechanistic implications of the study are: 1) TorsinA may not function as a hetero-hexamer (in contradiction to the prevailing model), and 2) the protein may be regulated by a redox mechanism via reversible formation of a disulfide bond between a pair of Cys residues located near the ATP binding site. Unfortunately, neither of these provocative proposals was proven definitively, and indeed the manuscript provides less mechanistic insight than would be ideal. Nevertheless, publication is warranted owing to: 1) the enzyme's importance (TorsinA has an intriguing, but poorly characterized role in nuclear membrane budding processes and it's mutations can cause dystonia), 2) the importance for the field of having a structural basis for understanding TorsinA and its interactions with activators (the results aren't entirely unexpected because donation of an activating Arg finger from the LULL1/LAP1 activators had been predicted, but having precise structures as an underpinning for further experimentation will be very helpful and could even guide the development of small molecules that could restore activator binding to the "deltaE" TorsinA mutants), and 3) the study will undoubtedly stimulate further work, including testing possible redox regulation and the possibility that TorsinA functions with activators as heterodimers (and not heterohexamers). The complex structure itself, together with the lack of conserved pore loop structures in TorsinA are quite consistent with this idea, but unfortunately the study isn't definitive because structure was determined in complex with a nanobody (which could have prevented oligomerization) and because the "backside" of Torsin A is conserved, so it remains possible that this surface interacts with other proteins (possibly even via homo-oligomerization with other TorsinA molecules). If a heterodimer is indeed the active minimal module, this would lend further credence to the concept that the TorsinA/LULL1 (and LAP1) complexes are "hybrids" between traditional ring AAA ATPases and G-protein/GAP systems.

The structures themselves are straightforward and are determined at high resolution. Ideally, the authors would test their provocative mechanistic proposals with additional biochemical analyses (beyond the rather rudimentary binding studies that are presented), but we are not requiring such studies owing to the absence of good in vitro assays or known substrates.

The discussion on the oligomeric state should be improved by including additional references. Specifically, it was previously published (Crit Rev Biochem Mol Biol. 2015;50(5):532-49) that the absence of the C-terminal subdomain in LULL1/LAP1 is difficult to reconcile with a closed, mixed ring conformation based on the requirement of the C-terminal four helix bundle in related Clp/HSP100 proteins. This should be explicitly stated, along with references reporting the requirement for the C-terminal subdomain in Clp ATPases. The authors should also include references reporting higher-order oligomeric assemblies of expressed Torsin, observed using Blue Native PAGE (Jungwirth et al. 2010; Goodchild et al. 2015). In this case, contributions of LAP1/LULL1 activators can be largely ignored. While these studies do not exclude the presence of additional cellular interaction partners, they do support the idea that Torsins can assemble into higher-order (>>2) homo-oligomeric assemblies, consistent with the conservation of possible Torsin-Torsin interfaces.

---

## [Author Response]

*The manuscript by Demircioglu et al. describes high resolution crystal structures of: 1) an "E/Q" (enzymatically inactive), but otherwise wild type TorsinA protein, in complex with its activator, LULL1, and 2) an analogous structure of the dystonia-causing "delta E" TorsinA protein in complex with LULL1. Strengths of the manuscript include the medical significance and scientific interest in TorsinA, and the importance for the field of knowing the precise structure of TorsinA and how it interacts with its activators. Two potentially important mechanistic implications of the study are: 1) TorsinA may not function as a hetero-hexamer (in contradiction to the prevailing model), and 2) the protein may be regulated by a redox mechanism via reversible formation of a disulfide bond between a pair of Cys residues located near the ATP binding site. Unfortunately, neither of these provocative proposals was proven definitively, and indeed the manuscript provides less mechanistic insight than would be ideal. Nevertheless, publication is warranted owing to: 1) the enzyme's importance (TorsinA has an intriguing, but poorly characterized role in nuclear membrane budding processes and it's mutations can cause dystonia), 2) the importance for the field of having a structural basis for understanding TorsinA and its interactions with activators (the results aren't entirely unexpected because donation of an activating Arg finger from the LULL1/LAP1 activators had been predicted, but having precise structures as an underpinning for further experimentation will be very helpful and could even guide the development of small molecules that could restore activator binding to the "deltaE" TorsinA mutants), and 3) the study will undoubtedly stimulate further work, including testing possible redox regulation and the possibility that TorsinA functions with activators as heterodimers (and not heterohexamers). The complex structure itself, together with the lack of conserved pore loop structures in TorsinA are quite consistent with this idea, but unfortunately the study isn't definitive because structure was determined in complex with a nanobody (which could have prevented oligomerization) and because the "backside" of Torsin A is conserved, so it remains possible that this surface interacts with other proteins (possibly even via homo-oligomerization with other TorsinA molecules). If a heterodimer is indeed the active minimal module, this would lend further credence to the concept that the TorsinA/LULL1 (and LAP1) complexes are "hybrids" between traditional ring AAA ATPases and G-protein/GAP systems.*

*The structures themselves are straightforward and are determined at high resolution. Ideally, the authors would test their provocative mechanistic proposals with additional biochemical analyses (beyond the rather rudimentary binding studies that are presented), but we are not requiring such studies owing to the absence of good in vitro assays or known substrates.*

*The discussion on the oligomeric state should be improved by including additional references. Specifically, it was previously published (Crit Rev Biochem Mol Biol. 2015;50([5]6):532-49) that the absence of the C-terminal subdomain in LULL1/LAP1 is difficult to reconcile with a closed, mixed ring conformation based on the requirement of the C-terminal four helix bundle in related Clp/HSP100 proteins. This should be explicitly stated, along with references reporting the requirement for the C-terminal subdomain in Clp ATPases. The authors should also include references reporting higher-order oligomeric assemblies of expressed Torsin, observed using Blue Native PAGE (Jungwirth et al. 2010; Goodchild et al. 2015). In this case, contributions of LAP1/LULL1 activators can be largely ignored. While these studies do not exclude the presence of additional cellular interaction partners, they do support the idea that Torsins can assemble into higher-order (>>2) homo-oligomeric assemblies, consistent with the conservation of possible Torsin-Torsin interfaces.*

We have revisited the existing literature and have rephrased this part of the discussion accordingly.